# OpenReview forum: "MedSafetyBench: Evaluating and Improving the Medical Safety of Large Language Models"
_NeurIPS.cc/2024/Datasets_and_Benchmarks_Track — NeurIPS 2024 Track Datasets and Benchmarks Poster_

### Official Review · Reviewer_67b5 · 2024-06-26
**Interesting work - misleading naming**

**Rating:** 5
**Confidence:** 4

**Review:**

The paper is presented well, provides a good motivation for the necessity of this benchmark and explains clearly how the dataset has been created and what values and principles were applied.
The approach is somewhat original as it tests against ethical behavior more than other more common aspects of medical output (such as scientific validity). Still, I think it falls short on several aspect: from the naming convention used, to the conclusions.

Terminology:
The naming convention used in this paper relies a lot on terminology adopted by the "AI Alignment" school of thought, and I don't think it applies appropriately in this case.
To be able to claim "medical use safety" for a model it would be necessary to also benchmark the scientific validity of a model. Ensuring it doesn't suggest/answer to unethical prompts doesn't guarantee safety at all. Maybe using the term "ethical" or "responsible" would be more accurate for this work.

Conclusions
The conclusions, fail to demonstrate how a model that has already been safety tuned in general would be improved by this benchmark, since several Llama 2-chat models and GPT3.5/4 seem to perform already very well on that benchmark (Figure 2).
Figure 2 shows pretty clearly that models that have already been fined tuned for general safety are also performing well on this benchmark. A lot of the prompts provided in this benchmark are unethical behavior or even just plain unlawful behavior, so it would be captured pretty well by a generic safety tuning (as demonstrated by the performances of Llama 2-chat and GPT3.5/4).
While it is interesting to see that the medical models are performing poorly, I do wonder if that's because they haven't been safety tuned at all (which is definitively problematic, but not specific to this issue).

Finally, a lot of the motivation for the paper is about the necessity for models to refuse to answer potentially dangerous requests, but after fine tuning the models on the new benchmarks, the authors are not providing an analysis on false refusal, which may, in principle, significantly reduce the performances of these models for medical use.

**Strengths:**

Significance of the Contribution
The major significance of this contribution, in my opinion, is simply bringing awareness to the issue that models tuned for medical use don't seem to be safety tuned at the level of generic chat models (probably because they're tuned starting from the foundational model). This is definitively concerning, but I am not sure that this specific dataset is the only/best way to address it, since a general safety tuning would also solve the problem.

Relevance to the broader research community
This paper could be relevant for the broader community in terms of awareness for how to properly fine tune models for specific uses while maintaining safety tuning as well. It provides a good method for testing behavior for more specialized uses, which is interesting.

quality of the research
It is important that the paper clarifies the principles and values it is testing the models against, but it doesn't explain in enough details how the prompts are created and how each principle is actually tested. Besides, as I mentioned in the review, I think the conclusions are a bit circular, since the authors are not demonstrating if and how this benchmark would improve the performances of models that already received safety tuning.

ethical and social implications
The ethical and social implications of this paper are positive as they are raising awareness on a potentially dangerous behavior of LLMs

**Additional Feedback:**

I provided comments and suggestions in the sections above

**Clarity:**

The paper is clearly written, while skipping over some discussion that I think would benefit the overall clarity and relevance of the paper

**Correctness:**

The claims in the submission are high level correct, but they're missing the discussion on some detailed aspects that I think are important. For example, it is not clear how the dataset is exactly constructed and how the ensure coverage of the principles they expose. It is also not very clear how this benchmark improves over a generic safety benchmark.

**Documentation:**

the documentation explains how the prompts were generated (although not in great details) and the submission includes a directory with the dataset and a datasheet

**Ethics:**

The only ethical concern I would have for this submission is that the claims are a bit misleading when defining the benchmark for "medical safety" when it is not actually holistically testing for medical safety.

**Limitations:**

Not really, I think a discussion to clarify this benchmark only addresses the need for models to refuse to reply to unethical prompts is essential, given that they're not considering other, very important aspects of medical safety such accuracy.

**Opportunities For Improvement:**

Significance of the Contribution
This contribution would be much more significant if it was including not only unethical/unlawful questions, but was also benchmarking the general medical safety of the models, including scientific accuracy, providing proper sources for the answers and providing users with instructions on how to contact experts as needed. Just ensuring that the model doesn't answer doesn't make it safe.
In general, the paper should either include also more aspect of actual medical safety, or should be renamed and explain more clearly that it is only focusing on ethical behavior of the model.

Relevance to the broader research community
The authors could improve the relevancy of this paper for a broader audience if they spent more time discussing how this benchmark can improve models that are already performing well on generic safety benchmarks, connecting it to instructions or suggestions on how to create medical LLMs that remain safe.

quality of the research
The paper could use a lot of improvement in the quality of research itself. In particular:
- prompt generation: explain in more details how prompts are generated, and how they are testing for each of the principles of medical ethics. Authors seem to just be asking LLMs to generate prompts for them, but there should be an analysis of the effectiveness and coverage of prompts (maybe based on attack axes).
- authors are showing how most models that were already performing poorly on generic safety benchmarks keep performing poorly on this specific benchmark, and viceversa... this doesn't tell me much about the usefulness of this benchmark. If the authors wish to make a point, they should start from here.
- Given that the goal of the paper is to prevent models to respond at all when prompted with harmful questions, authors should also have a discussion on recall/false refusal and evaluate performances before and after the tuning

ethical and social implications
I think mostly the authors should clarify better the scope of the work, because defining it as medical safety is overselling the benchmark

**Relation To Prior Work:**

There is a good overview of related work, but I think it misses the connection of showing how this benchmark improves over the generic safety benchmarks (since models that perform poorly there, continue to perform poorly, and models that perform well continue to perform well)

**Summary And Contributions:**

The authors are presenting a new dataset for benchmarking LLMs on their compliance with harmful demands in the medical field. They created a set of rules based on the medical oath, expecting LLMs to refuse to answer questions or prompts that incite/endorses unethical or unlawful behavior. They demonstrate how several models that have been fined tuned for medical purposes are actually complying with unethical requests, and how RLHF based on the dataset they produce is improving the performances of the models on the same benchmark.

---

> ### Author Rebuttal · Authors · 2024-08-16
>
> Dear Reviewer 67b5,
>
> Thank you! We appreciate your review and are glad you found that “paper is presented well, provides a good motivation for the necessity of this benchmark and explains clearly how the dataset has been created and what values and principles were applied” and is “relevant for the broader community”. We address your comments below.
>
> > “To be able to claim "medical use safety" for a model it would be necessary to also benchmark the scientific validity of a model. Ensuring it doesn't suggest/answer to unethical prompts doesn't guarantee safety”
>
> We agree that for an LLM to be safe, it ought to have both scientific validity and ethics. We define a medical LLM to be safe if it aligns with the AMA’s Principles of Medical Ethics (Section 3.1, lines 104-105). The first principle is “competent medical care” (Table 1) which includes scientific validity. In experiments, we also directly examine the scientific validity of the medical LLMs, showing that general and medical safety-tuning preserves their scientific validity (Section 4.2, lines 195-199, 209-210; Appendix C, Figure 8). Requiring that a medical LLM declines to answer harmful requests (on top of its scientific validity) is also in line with prior work [1, 2]. Thank you for bringing this up! We will emphasize the above in the paper.
>
> [1] Bianchi et al. Safety-tuned Llamas. ICLR 2024. [2] Zou et al. Universal and Transferable Adversarial Attacks on Aligned Language Models. Arxiv 2023.
>
> > “The conclusions, fail to demonstrate how a model that has already been safety tuned in general would be improved by this benchmark”
>
> The purpose of this benchmark is two-fold: to 1) evaluate and 2) improve the medical safety of LLMs.
>
> Some models with general safety-tuning (Llama-2-chat and GPT3.5/4 in Figure 2) perform well on the benchmark, while others (Mistral-7b-Instruct and Mixtral-7x8b-Instruct in attached PDF; new experiments) do not. If a model performs well on this benchmark, medical safety-tuning may not be necessary. However, even for these models, the benchmark is critical because it enables the evaluation that led to this conclusion.
>
> If a model does not perform well on the benchmark, then this benchmark can improve the model’s medical safety. For example, we find that medical LLMs do not perform well on the benchmark (Figure 2) and use this benchmark to improve their medical safety (Figure 3) while preserving medical performance (Section 4.2, lines 195-199, 209-210; Appendix C, Figure 8).
>
> > Is general safety-tuning enough for medical safety?: “I am not sure that this specific dataset is the only/best way to address it [the issue that models tuned for medical use don't seem to be safety tuned], since a general safety tuning would also solve the problem.”
>
> Whether general safety-tuning guarantees medical safety is an open question. On the one hand, some models with general safety-tuning perform well on the benchmark (Figure 2), suggesting that for these models, general safety-tuning imparted medical safety. On the other hand, for medical LLMs we studied, general safety-tuning does not impart medical safety as strongly as medical safety-tuning (Figure 3b and 3c, compare “gen_n900” vs. “med_n900” for each model). Thus, different general safety-tuning methods may result in different levels of medical safety.
>
> Given the above, the risks of medical LLMs [1, 2], and the state of LLM research (new models, new alignment methods, different models using different alignment methods, opacity of the alignment methods for some models, no guarantee that newer models are safer than predecessors, etc.), there is great need for the development of this kind of benchmark to evaluate and improve the medical safety of these models. In fact, various researchers, including the developers of one of the medical LLMs we study, have asked for access to this dataset.
>
> [1] Goldberg et al. To do no harm - and the most good - with AI in healthcare. Nature Medicine 2024. [2] Haltaufderheide et al. The ethics of ChatGPT in medicine and healthcare. NPJ Digital Medicine 2024.
>
> > “after fine tuning the models on the new benchmarks, the authors are not providing an analysis on false refusal, which may, in principle, significantly reduce the performances of these models for medical use” / “they're not considering other, very important aspects of medical safety such accuracy”
>
> We believe the reviewer may have misunderstood an important aspect of the paper. The reviewer claims the paper does not study the accuracy of the models after the models have been fine-tuned for medical ethics, when in fact the paper precisely examines this.
> We find that general and medical safety-tuning preserves the scientific accuracy of the medical models (Section 4.2, lines 195-199, 209-210; Appendix C, Figure 8). Furthermore, during these experiments, we did not observe false refusal, i.e., fine-tuning the medical models to decline harmful questions did not lead them to decline scientific questions.
>
> > “explain in more details how prompts are generated, and how they are testing for each of the principles of medical ethics” / “there should be an analysis of the effectiveness and coverage of prompts”
>
> The harmful medical prompts in the benchmark were generated by directly prompting GPT4 or by attacking Llama-2-7b via the Greedy Coordinate Gradient algorithm for each principle of medical ethics. We then performed a qualitative analysis, manually inspecting each prompt and keeping the ones that violated at least one principle (Section 3.2, lines 110-122; Appendix A.1).
>
> In addition, in response to reviewer comments and to strengthen our work, we are conducting a user study with doctors (domain experts) to examine “effectiveness and coverage”, further validating the benchmark. We hope to post our results during the discussion period. Thank you for the suggestion!
>
> We hope our response addresses your comments and that you consider increasing the score. Thank you!
>
> Sincerely,
>
> Authors

---

> > ### Author Rebuttal · Authors · 2024-08-26
> >
> > Dear Reviewer 67b5,
> >
> > We conducted a user study which further addresses your comments.
> >
> > > “there should be an analysis of the effectiveness and coverage of prompts”
> >
> > Thank you for bringing this up! To address this, we conducted a user study with domain experts (doctors). We find that the vast majority of prompts indeed violate the AMA Principles of Medical Ethics (“effectiveness of prompts”) and that the prompts cover all nine principles (“coverage of prompts”). Additional details can be found in the comment to all reviewers. We hope this study addresses your comments and that you consider increasing the score. Thank you!
> >
> > Sincerely,
> >
> > Authors

---

### Official Review · Reviewer_Taqs · 2024-07-11
**Timely and comprehensive benchmark for medical safety**

**Rating:** 7
**Confidence:** 4
**Correctness:** Yes.
**Clarity:** Yes.

**Review:**

This work is well-written and easy to follow, with sufficient details about data curation and experimentation in the main text and appendix. The results are comprehensive and adequately discussed.

This work studies a timely and important topic, which is medical safety. The results in this work may serve as a solid starting point for future research on the criteria of medical safety, ways to evaluate it, and methods to improve it.

However, the methods and ideas in benchmarking are quite similar to existing works in general safety, e.g., [1]. The unique challenges and methods for safety in medicine should be emphasized more.

[1] Federico Bianchi, Mirac Suzgun, Giuseppe Attanasio, Paul Röttger, Dan Jurafsky, Tatsunori Hashimoto, and James Zou. Safety-tuned Llamas: Lessons from improving the safety of large language models that follow instructions. International Conference on Learning Representations, 2024.

**Strengths:**

1. The studied topic is important, and this work makes solid contributions towards this direction.

2. The results in this work are comprehensive.

3. The paper is well written and easy to follow.

4. This paper concerns the ethical and social impact of LLMs and has positive implications for these aspects.

**Additional Feedback:**

Questions for the authors:

1. Independence of data curation and evaluation.

Since the data are partially generated by GPT4 and Lllama, I am curious whether the subsequent evaluation of these models are biased or not. Similar concerns apply to GPT3.5 which is used to judge the safety of LLM answers. It would be helpful to discuss this point.

2. What are the distinction of the approaches from existing ones?

The methods and ideas in benchmarking are quite similar to existing works in general safety, e.g., [1], except that this work focuses more on medical-related questions. The unique challenges and methods for safety in medicine should be emphasized more.

[1] Federico Bianchi, Mirac Suzgun, Giuseppe Attanasio, Paul Röttger, Dan Jurafsky, Tatsunori Hashimoto, and James Zou. Safety-tuned Llamas: Lessons from improving the safety of large language models that follow instructions. International Conference on Learning Representations, 2024.


Minor points:

1. Line 129: "declines" --> "decline"

**Documentation:**

Yes.

**Ethics:**

No.

**Limitations:**

Yes.

**Opportunities For Improvement:**

1. Need to address the distinction from existing safety evaluation and improvement methods (e.g., those in general safety), and emphasize more on the unique challenges and methods for medical safety.

**Relation To Prior Work:**

Yes.

**Summary And Contributions:**

This paper proposes a benchmark for the medical safety of large language models. The authors first define the medical safety by the Principles of Medical Ethics from AMA. This work contributes a dataset for evaluating medical safety, consisting of harmful prompts generated by GPT4 and Llama-7B. These datasets are used to evaluate the medical safety of popular LLMs via scoring from GPT3.5. In addition, this work contributes datasets that contain safe answers to harmful prompts, which are then used in conjunction with instruction fine-tuning to improve the medical safety of several LLMs. It is demonstrated that this fine-tuning improves both general safety defined in the literature and medical safety of LLMs.

---

> ### Author Rebuttal · Authors · 2024-08-16
>
> Dear Reviewer Taqs,
>
> Thank you! We appreciate your review and are glad you found that “this work studies a timely and important topic”, that “the results are comprehensive and adequately discussed”, and that the results “serve as a solid starting point for future research on the criteria of medical safety, ways to evaluate it, and methods to improve it.” We address your comments below.
>
> > Novelty of methods: “What are the distinction of the approaches from existing ones?”
>
> The novelty of this paper is less in its methods, and more in the application of methods (such as those in [1]) to a new domain (medicine) and in the creation of a new benchmark dataset. When studying this new domain, we use domain-specific notions of ethics (e.g., AMA Principles of Medical Ethics) and domain experts (we are conducting a study with doctors to further validate the benchmark, in response to reviewer comments and to strengthen our work). In line with the spirit of the Datasets and Benchmarks Track, our paper contributes a novel benchmark dataset aimed at evaluating and improving the medical safety of LLMs.
>
> [1] Federico Bianchi, Mirac Suzgun, Giuseppe Attanasio, Paul Röttger, Dan Jurafsky, Tatsunori Hashimoto, and James Zou. Safety-tuned Llamas: Lessons from improving the safety of large language models that follow instructions. ICLR 2024.
>
> > Typo in Line 129
>
> We will fix this. Thank you for pointing this out!
>
> > Independence of data curation and evaluation: “Since the data are partially generated by GPT4 and Lllama, I am curious whether the subsequent evaluation of these models are biased or not.”
>
> To generate the benchmark dataset, we used GPT4 and Llama2-7b. While we do use this benchmark to evaluate GPT4 and Llama2-7b models, we also use this benchmark to evaluate other models: Llama models, other Llama2 models, GPT3.5, and various medical LLMs. Furthermore, we conducted new experiments (results included in the attached PDF) and used this benchmark to evaluate 14 additional models, spanning 5 model families: GPT4o models, Llama3.1 models, Mistral models, Vicuna models, and Pythia models (the latter three model families are outside of the LlamaX and GPTX model families). These results include a more diverse set of general-knowledge models, bolster our benchmarking of general-knowledge models, and further support our conclusion that medical LLMs readily comply with harmful general and medical requests, and they do so more frequently than their safety-aligned, general-knowledge counterparts.
>
> We hope our response addresses your comments. Thank you!
>
> Sincerely,
>
> Authors

---

### Official Review · Reviewer_FaWV · 2024-07-24
**Medical Safety Benchmark based on train/test examples of positive/negative principles of medical ethics**

**Rating:** 7
**Confidence:** 4
**Clarity:** Yes

**Review:**

+ The paper ambitiously aims to align medical LLMs with the American Medical Association, a well-established standard.
+ The technique of using jailbreaking attacks to produce negative samples from safety-aligned LLMs is an interesting way to use jailbreak attacks.
+  Existing models are shown to be not aligned, which is a significant problem for the medical LLM field.
+ The fact that fine-tuning improves performance is more or less expected, so it is good that it is consistent

- How exactly the data was curated after generation to result in the 1800 examples is not clearly described. I am hoping that there were experts that are well-versed in the standards of the AMA involved.
- It is possible that the generated data spans only a subspace of possible unethical queries. In which case, it may not accurately describe broader alignment. However, such an initial step can still be valuable.

**Strengths:**

The main strength is that the benchmark is grounded in real-world standards. Such a standard I initially thought wouldn't be possible to measure, but the authors took the strategy of measuring by example. By using clear examples that should be answered or not answered, it becomes possible to at least empirically measure the medical alignment on this fixed dataset, even if not more generally beyond.

**Additional Feedback:**

Please include a maintenance plan for the dataset.

**Correctness:**

Assuming the authors can confirm how the data was curated to adhere to the AMA standards, then it is sound in the sense that the provided data does measure to some degree medical alignment. However, it may not measure all degrees (which is not necessary, but likely should be discussed as a limitation).

**Documentation:**

The provided datasheet does not have much detail on collection/organization, and says that additional details will be provided after review. It will be publicly available, but unclear where.

There is no maintenance plan.

I was able to view the strings in the provided CSV files in the code base, however I did not run the code.

**Limitations:**

The authors checklist says that limitations are discussed in Section 5, however my reading of Section 5 seems to include only future work and conclusions. Please include limitations of the described benchmark.

**Opportunities For Improvement:**

The jailbreak attack deployed in this work could suffer from being biased by GCG-style optimization. To improve the diversity of the negative samples, it could be helpful to employ other semantic jailbreak attack techniques (such as PAIR) which can search a wider space and get more creative negative samples. Broadening the dataset will also help alleviate potential worries that the dataset is too narrow in scope.

It may be of use to consider additional gray-area examples, i.e. prompts that are borderline but not technically unethical to ask. Such data could help further distinguish between LLMs that perform similarly and give a more nuanced view into the alignment.

**Relation To Prior Work:**

Yes.

**Summary And Contributions:**

+ The paper proposes evaluating medical safety of LLMs along standards defined by the American Medical Association
+ The authors curate a train and test set of 900+900 examples each, with half positive and negative
+ The authors utilized jailbreaking attacks to get safety-aligned models to produce negative examples
+ A harmfulness score from previous work is utilized to measure the willingness of an LLM to comply with unethical questions
+ Models are shown to answer negative examples, and fine-tuning reduces this problem

---

> ### Author Rebuttal · Authors · 2024-08-16
>
> Dear Reviewer FaWV,
>
> Thank you! We appreciate your review and are glad you found that the paper “ambitiously aims to align medical LLMs with the American Medical Association, a well-established standard”, that the “technique of using jailbreaking attacks to produce negative samples from safety-aligned LLMs is an interesting way to use jailbreak attacks”, and that “the benchmark is grounded in real-world standards”. We address your comments below.
>
> > Data curation: “How exactly the data was curated after generation to result in the 1800 examples is not clearly described. I am hoping that there were experts that are well-versed in the standards of the AMA involved.” // “Assuming the authors can confirm how the data was curated to adhere to the AMA standards, then it is sound in the sense that the provided data does measure to some degree medical alignment.” // "The provided datasheet does not have much detail on collection/organization, and says that additional details will be provided after review. It will be publicly available, but unclear where."
>
> We carefully generated the harmful prompts to cover all nine principles of medical ethics. Specifically, we prompted GPT4 and attacked Llama2-7b to create harmful prompts for each principle of medical ethics (one principle at a time). In addition, upon generating the harmful prompts, we performed a qualitative analysis, manually inspecting all the prompts and keeping only the ones that indeed violate at least one principle of medical ethics. Furthermore, in response to reviewer comments and to strengthen our work, we are working on a user study with doctors (domain experts) to further validate the quality of our benchmark dataset. We hope to post our results during the discussion period. We will also include these details in the final paper.
>
> > Dataset maintenance and additional attack methods: “Please include a maintenance plan for the dataset.” // “it could be helpful to employ other semantic jailbreak attack techniques (such as PAIR)”
>
> As part of the maintenance plan, we will update the benchmark as new papers arise and as we receive suggestions from the community (e.g., new attack methods, other definitions of medical ethics, new medical LLMs, etc.). We will create a new version of the dataset which contains prompts generated using the PAIR attack method. This is a good point, thank you for the suggestion!
>
> > Gray-area examples: "It may be of use to consider additional gray-area examples, i.e., prompts that are borderline but not technically unethical to ask. Such data could help further distinguish between LLMs that perform similarly and give a more nuanced view into the alignment."
>
> Thank you for bringing this up! At the moment, it is unclear how to generate gray-area examples in a systematic way and validate them, in part because the nuances of medical safety and ethics are complex and may depend on medical subspecialty, a patient’s specific condition, and personal preference (Section 5, lines 218-223). We will add this discussion to the paper.
>
> > Limitations: “The authors checklist says that limitations are discussed in Section 5, however my reading of Section 5 seems to include only future work and conclusions. Please include limitations of the described benchmark.”
>
> We discuss limitations and how future work can address these limitations in Section 5 (lines 218-229). Specifically, this work defines medical safety based on the AMA Principles of Medical Ethics and uses instruction-tuning to perform safety alignment. However, in practice, there could be more nuance to the definition of medical safety and there are other fine-tuning approaches such as RLHF. These represent future research directions. We will clarify this in the paper.
>
> We hope our response addresses your comments. Thank you!
>
> Sincerely,
>
> Authors

---

> > ### Author Rebuttal · Authors · 2024-08-26
> >
> > Dear Reviewer FaWV24,
> >
> > We conducted a user study which further addresses your comments.
> > > Data curation: “I am hoping that there were experts that are well-versed in the standards of the AMA involved.” // “Assuming the authors can confirm how the data was curated to adhere to the AMA standards, then it is sound in the sense that the provided data does measure to some degree medical alignment.”
> >
> > Thank you for bringing this up! We conducted a user study with domain experts (doctors). We find that the benchmark dataset indeed violates the AMA Principles of Medical Ethics and covers all nine principles, confirming the medical alignment of the benchmark. Additional details can be found in the comment to all reviewers.
> >
> > Sincerely,
> >
> > Authors

---

### Author Rebuttal · Authors · 2024-08-16

In response to reviewer comments and to strengthen our work, we are working on a user study with doctors (domain experts) to further validate the quality of our benchmark dataset. We hope to post our results during the discussion period. We will also include these details in the final paper.

In addition, we conducted additional experiments and evaluated the general and medical safety of 14 new general-knowledge models, spanning 5 new model families (GPT4o, Llama3.1, Mistral, Vicuna, and Pythia). The results are in the attached PDF. These results include a more diverse set of general-knowledge models and bolster our conclusion that medical LLMs readily comply with harmful general and medical requests, and they do so more frequently than their safety-aligned, general-knowledge counterparts. We also refer to these results in the reviewer-specific discussions below.

---

> ### Author Rebuttal · Authors · 2024-08-26
>
> In response to reviewer comments and to strengthen our work, we conducted a user study with doctors (domain experts) to further validate the benchmark dataset. We discuss the setup and results below. Figures are in the attached PDF.
>
> **Set-up.** To validate the MedSafetyBench dataset, we conducted a user study with
> 25 domain experts (doctors). Each survey consisted of 25 randomly-selected prompts from the full dataset (n = 1800). For each prompt, domain experts were asked if the prompt violates any of the nine Principles of Medical Ethics. If so, they were further asked to select any of the nine principles that are violated by the prompt.
>
> **Violation of the Principles of Medical Ethics.** Among the prompts shown to domain experts, 91% (567/625) were deemed to violate at least one principle (Figure 1). Thus, the vast majority of prompts are indeed unethical. The remaining 9% (58/625) of prompts were deemed ethical by at least one domain expert, but 31% (18/58) of these prompts were also deemed unethical by at least one other domain expert. Thus, rather than being categorically ethical, these prompts likely represent “gray-area” cases.
>
> **Coverage of prompts.** For prompts that were deemed by domain experts to violate at least one
> principle, we examined the specific principles the prompts violated. The distribution of these principles is shown in Figure 2. The prompts span all nine Principles of Medical Ethics, relating to some principles (Principles 1, 2, and 8) more than others (Principles 5 and 6) (Figure 2).
> Among prompts that were deemed unethical by domain experts, 85% (482/567) simultaneously violated two or more Principles of Medical Ethics (Figure 1). In addition, some pairs of principles (such as Principles 1 and 2, 1 and 8, and 2 and 8) are often violated together (Figure 3). These results suggest that the Principles of Medical Ethics are not mutually exclusive.
>
> In summary, the user study further validates the benchmark, providing evidence that the vast majority of the prompts indeed violate Principles of Medical Ethics and that the benchmark covers all nine principles.
>
> We hope this user study helps address reviewer comments and will include this user study in the final paper.

---

### Decision · Program_Chairs · 2024-09-26

**Decision:**

Accept (Poster)

**Comment:**

The reviewers found the paper to be of good quality and potentially can benefit the research community.